# VARAN: VARIATIONAL INFERENCE FOR SELF-SUPERVISED SPEECH MODELS FINE-TUNING ON DOWNSTREAM TASKS

## ABSTRACT

Despite the growing interest in self-supervised speech models, recent research has primarily focused on modifying upstream model architectures and pretraining techniques, with less attention given to how features from self-supervised models are used. In this paper, we explore the use of variational inference to enhance the performance of self-supervised audio models in downstream tasks. We hypothesize that adaptively reweighting the outputs of the model layers is crucial to improving performance on these tasks. We extensively evaluate our method alongside widely used baselines, demonstrating that understanding sample-specific information is essential for improved performance on several tasks. Our proposed method surpasses existing approaches and generalizes to various speech tasks, including automatic speech recognition, speaker verification, and emotion recognition. Finally, we analyze our method to provide deeper insight into the importance of our modifications.

## 1 INTRODUCTION

Self-supervised learning (SSL) approaches for speech have emerged to address the need for a cost-effective way to achieve high performance with minimal annotated data across a wide range of downstream tasks. Modern SSL audio models are trained on massive unlabeled audio corpora and show high performance while fine-tuning for downstream tasks with as little as 1 hour of labeled audio (Chen et al., 2022). Moreover, SSL models can serve as neural feature extractors capable of replacing traditional audio representations such as raw waveforms or spectrograms, and consistently outperform models trained with these classical features (Boigne et al., 2020; Peng et al., 2022).

The most widely used SSL models today include wav2vec 2.0 (Baevski et al., 2020), HuBERT (Hsu et al., 2021), WavLM (Chen et al., 2022), and Data2Vec (Baevski et al., 2022). Architecturally, these models consist of a CNN feature extractor, which is typically kept frozen during fine-tuning on downstream tasks Stafylakis et al. (2022), and a Transformer's (Vaswani et al., 2017) encoder, which can either be kept frozen or trained during fine-tuning (wen Yang et al., 2021a). There are two most common methods for using the encoders' outputs to make predictions for downstream tasks: taking the output from the last layer or learning weights for all Transformer layers and using a weighted output (wen Yang et al., 2021a).

The motivation behind collapsing self-supervised model output into a single feature using learned weights is based on the observation that different layers of SSL models contain varied information, which may be required depending on the task. One way to conduct layer importance analysis is by comparing the absolute values of the learned weight parameters. Using this method, Chen et al. (2022) demonstrated that the initial layers obtain higher weights than the later layers after training on speaker-dependent tasks, indicating that they contain information essential for those tasks. The last layers of SSL models contain more semantic and phonetic information (Pasad et al., 2023), which is typically required for Automatic Speech Recognition (ASR) tasks, as demonstrated in Pasad et al. (2023) through canonical correlation analysis.

However, while using a weighted sum with weights learned jointly with the downstream model and analyzing them for task-specific layer importance of the SSL layers is a widely used and straightforward approach, it has several shortcomings. Firstly, this approach does not account for sample

variation and its possible dependence on information from different SSL layers. Secondly, Yang et al. (2024) showed that the performance of individual layers of SSL models does not correlate with the learned weights.

In this paper, we address these drawbacks and propose a layer aggregation method for SSL models based on variational inference. The proposed method, called **VARAN**, predicts the weights for each individual layer based on the particular sample. We demonstrate that the proposed method generalizes well across various tasks and SSL model architectures. We compare the proposed method to state-of-the-art layer-aggregating methods, including weighted sum and layerwise attention pooling, and show that our method outperforms existing layer-aggregation approaches on most tasks. Additionally, we conduct an analysis of the proposed method.

Overall, our contributions can be described as follows:

1. We propose an algorithm to fine-tune self-supervised speech models using variational inference.

2. We demonstrate that the proposed method generalizes well across various downstream tasks and upstream backbones.

3. Based on the obtained results, we provide a layer-wise analysis and an ablation study of the proposed method's robustness to its hyperparameters across various tasks.

## 2 BACKGROUND

In this section, we take a look at the layer-aggregation methods used for fine-tuning self-supervised speech models to downstream tasks.

### 2.1 SELF-SUPERVISED SPEECH MODELS NOTATION

Usually, a self-supervised speech model (upstream encoder $E$) takes a raw audio signal $x \in \mathbb{R}^n$, where $n$ is the number of samples in the raw audio signal, and produces an encoded output $E(x) = h \in \mathbb{R}^{[N \times T \times d]}$, where $N$, $T$, and $d$ are the number of layers in the Transformer, the encoded sequence length, and the Transformer hidden dimension, respectively.

The dimension of the input to the downstream model depends on the task. For phoneme recognition (PR) or automatic speech recognition (ASR) tasks, the input hidden vector needs to be $h \in \mathbb{R}^{[T,d]}$. For classification tasks, the input hidden vector needs to be $h \in \mathbb{R}^d$, obtained by applying mean or attention pooling across the sequence length.

To accumulate the results across layers, we can either take the input of the last Transformer layer $h_N$, or use a weighted sum across all layers: $h = \sum_{i=1}^{N} w_i \cdot h_i$, where $w$ is a softmax of learnable weights parameter.

### 2.2 MULTI-HEAD FACTORIZED ATTENTIVE POOLING

In this work, we explore enhanced techniques to fine-tune pretrained self-supervised speech model architectures by reweighting Transformer Encoder outputs. The closest work to ours is Peng et al. (2022), where the authors introduced Multi-Head Factorized Attentive Pooling (MHFA).

The intuition behind MHFA is that the model learns to produce values $\mathbf{V} = \left( \sum_{i=1}^{N} w_i^v \mathbf{h}_i \right) \mathbf{S}^v$, $\mathbf{V} \in \mathbb{R}^{[T \times \hat{d}]}$ that contain only speaker-discriminative information, while the keys $\mathbf{K} = \left( \sum_{i=1}^{N} w_i^k \mathbf{h}_i \right) \mathbf{S}^k$, $\mathbf{K} \in \mathbb{R}^{[T \times \hat{d}]}$ may contain phonetic information. Here, $\mathbf{S}^k$ and $\mathbf{S}^v \in \mathbb{R}^{[d \times \hat{d}]}$ are linear projections, with $\hat{d}$ being an attention hidden dimension. Queries are $Q \in \mathbb{R}^{[\hat{d} \times H]}$, where $H$ is the number of attention heads. Attention weights are computed as: $\mathbf{A} = \text{Softmax}(\mathbf{KQ})$. Each attention head can aggregate information from a specific set of phonetic units. At the frame level, values are aggregated using multi-head attention: $\mathbf{c}_{H_i} = \sum_{t=1}^{T} \mathbf{A}_{H_i,t} \mathbf{V}_t$, where $H_i$ corresponds to the output of the $i$-th attention head. Finally, the resulting values $\mathbf{c}_{H_i}$ are concatenated along the hidden dimension $\hat{d}$.

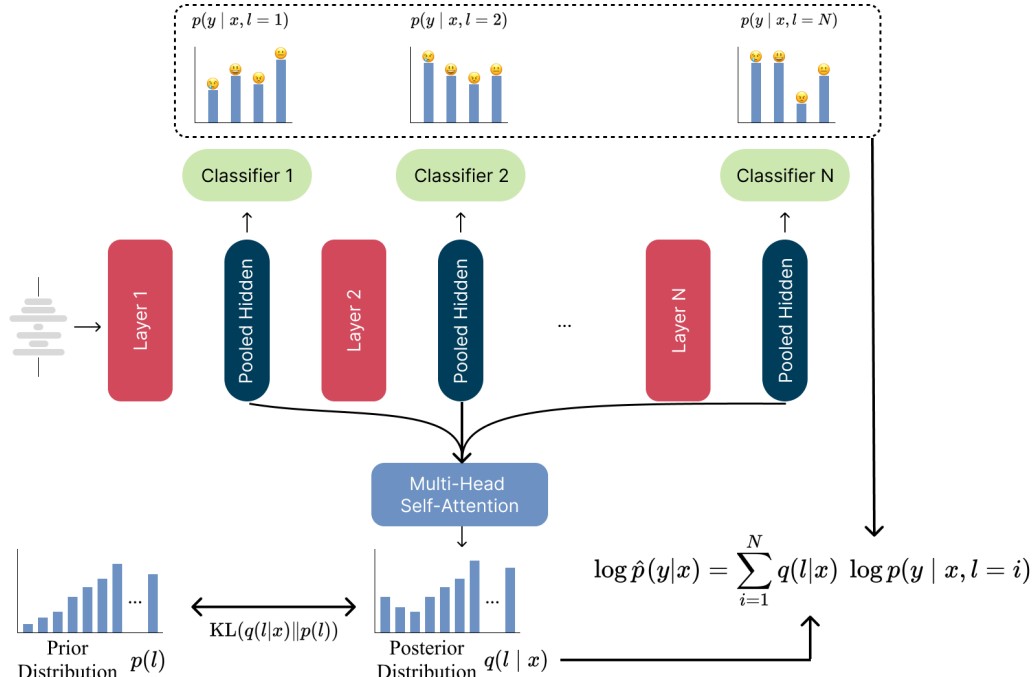

Figure 1: Overview of the **VARAN** architecture. The model consists of classifiers $p(y \mid x, l = i)$ for every layer index $i$, which yields distribution on $y$ target variable and layer distribution predictors $q(l|x)$, which aggregate information from each layer. After obtaining all distributions, log probabilities of the $y$ are obtained. See Section 3.2 for more details.

## 3 METHOD

In this section, we introduce the proposed method for aggregating the outputs of self-supervised speech models and explain how it can be used for training and inferring various downstream tasks. We begin with the overall intuition behind using Variational Inference for layer aggregation, followed by a detailed explanation of the choice of prior distribution and the prediction of the posterior distribution.

### 3.1 FORMULATING THE DOWNSTREAM TASK IN TERMS OF VARIATIONAL INFERENCE

To view any downstream task for SSL models from the perspective of variational inference, we introduce the intuition behind the approach. We assume that for any downstream task, our goal is to predict the target $y$ using input data $x$. One could use a simple linear classifier on top of the last hidden layer, but we found this setup suboptimal, as we need a different layer to achieve the best performance for different tasks and even different samples. Choosing the right layer or combination of layers is challenging, as the training data does not have the optimal layer distribution. However, we can learn it with the help of variational inference. In our setup, we have a classifier on top of each backbone's layer $p_\theta(y \mid x, l = i)$ and layer classifier $q_\theta(l|x)$, where $l$ is a discrete random variable and $i$ is a layer index. To maximize $p(y|x)$, we can use the following objective:

$$L(\theta) = -\mathbb{E}_{q_\theta(l|x)} \log p_\theta(y \mid x, l) + \mathrm{KL}\left(q_\theta(l \mid x)\|p(l)\right) \geq -\log p(y \mid x),$$

where $p(l)$ is a prior distribution on layer index.

Inspired by $\beta$-VAE (Higgins et al., 2017), we use a $\beta$ scaling factor as the regularization term. With this modification, our objective can be written as:

$$L(\theta) = -\mathbb{E}_{q_\theta(l|x)} \log p_\theta(y \mid x, l) + \beta \, \mathrm{KL}\left(q_\theta(l \mid x)\|p(l)\right). \tag{1}$$

You can find more details on derivation in Appendix A.

## 3.2 PARAMETRIZATION OF THE DISTRIBUTIONS

An overview of the proposed method can be found in Figure 1.

We start with the classifiers $p_\phi(y \mid x, l = i)$. First, for tasks that require temporal dimension reduction, we obtain the pooled hidden state $h$ via average pooling on the temporal dimension, where $h = E(x)$. The pooled hidden state is passed into a projector with ReLU non-linearity and then to a linear classifier, i.e., for the WavLM (Chen et al., 2022) Large model, this results in 24 probability distributions on $y$. To accumulate these distributions, we use a layer distribution predictor $q_\theta(l \mid x)$. The layer distribution predictor is a multi-head self-attention mechanism (MHSA).

The input to the MHSA consists of hidden states from the Transformer Encoder blocks. Formally, this is a list with a size equal to the number of layers, $L$, where each element is a tensor $\mathbf{l} \in \mathbb{R}^{b \times s \times h}$, with dimensions corresponding to batch size, sequence length, and hidden size. We then apply mean pooling along the sequence length and concatenate the results into a single tensor, resulting in a tensor $\mathbf{l}_{\text{pooled}} \in \mathbb{R}^{b \times L \times h}$ This tensor serves as input to the standard MHSA mechanism, functioning as a feature mixer between layers. Finally, we apply a linear layer to the last dimension to obtain a tensor $\mathbf{o} \in \mathbb{R}^{b \times L \times 1}$. We then perform a squeeze operation to remove the singleton dimension and apply softmax function along the number of layers dimension to produce an individual posterior distribution for each sample in the batch, $\mathbf{p} \in \mathbb{R}^{b \times L}$

Note that we are minimizing negative log-likelihood, and it is crucial to marginalize log-probabilities $\log p_\phi(y \mid x, l)$ over layer distribution, instead of marginalizing $p_\phi(y \mid x, l)$ and then taking $\log(\cdot)$ over it.

## 3.3 PRIOR DISTRIBUTION CHOICE

Selecting a prior distribution is crucial for achieving better performance, as discussed in Section 5.1. When selecting the prior, we can leverage domain-specific knowledge about the task. Previous work (Yang et al., 2024; Pasad et al., 2023) has shown that different features correspond to different layers: low-level audio features tend to be found in the early layers, while high-level features, such as phonemes, appear in the later layers. Thus, for tasks that are highly dependent on semantic information, like ASR or PR, we can use a prior distribution that assigns more weight to the later layers, such as a reversed geometric distribution. However, for most tasks, we do not have prior knowledge, and the layer importance for a specific task may vary depending on the choice of the backbone model. Therefore, in our setup, we tend to select the best prior distribution using hyperparameter search. We found that the scaled non-central $\chi^2$ distribution can be easily adapted to most tasks, as its shape aligns with our intuition of the task's prior distribution. We vary the non-centrality parameter $\lambda$ to shift the distribution toward different layers. See Appendix C for more details on the parametrization.

## 4 EXPERIMENTS

In this section, we describe the experimental setup and the results obtained by comparing the layer-aggregation methods discussed in Section 2. In addition to the described methods, we also introduce the Weighted Hiddens layer aggregation method, which is similar to the Weighted Sum but uses weights predicted by the MHSA mechanism employed for distribution prediction in the VARAN architecture. To examine the VARAN layer aggregation method, we selected the following three tasks: automatic speech recognition (ASR), speech emotion recognition (SER), and speaker verification (SV).

## 4.1 SETUP

To properly validate the ability of our method to outperform other layer aggregation techniques, we decided to unfreeze all backbones in this section. The setup described in the SUPERB benchmark (wen Yang et al., 2021b) is designed to validate encoder representations by freezing the Transformer encoder layers. However, in our work, we focus on comparing fine-tuning methods.

In all of our experiments, we unfreeze the encoder weights while keeping the CNN feature extractor parameters frozen. We find that this setup produces better results across all methods and datasets. All experiments except speaker verification are conducted with WavLM Large (Chen et al., 2022) [1] and Data2Vec Large (Baevski et al., 2022) models with the hyperparameters used during pre-training. For each layer aggregation method, we use gird search to find the optimal hyperparameters. A detailed description, including the hyperparameter grid and the hyperparameters chosen for VARAN's prior distribution for each task, can be found in Appendix E. Extended results can be found in Appendix F.

## 4.2 ASR

For the automatic speech recognition task, we compare layer-aggregation methods using the LibriSpeech-100 training dataset (Panayotov et al., 2015). We report results on test-clean and test-other sets. All models are trained with linear projectors and classifiers. We utilize Beam Search decoding without a language model since its ability to align text may introduce bias in evalution.

Our results are presented in Table 1. It is evident that VARAN outperforms other models on the test-clean set, while the Weighted Sum and Weighted Hiddens methods achieve lower WER scores on the test-other set only for WavLM backbone. To show the generality of VARAN, we also provide experiments with other languages, see Appendix F

Table 1: (Updated.) WER for different layer aggregation methods trained on Speech Recognition task. All models are trained with 5 seeds on LibriSpeech-100 training corpus (Panayotov et al., 2015). Varan method performs best except WavLM backbone on "test-other" evaluation set, where our method performs on par with weighted sum and weighted hiddens. See Section 4.2 for more details.

| Method | Backbone | WER clean ($\downarrow$) | WER other ($\downarrow$) |
|---|---|---|---|
| Last Layer | | $3.28 \pm 0.05$ | $6.75 \pm 0.10$ |
| Weighted Sum | WavLM Large | $3.21 \pm 0.05$ | $6.39 \pm 0.06$ |
| Weighted Hiddens | | $3.22 \pm 0.03$ | $6.39 \pm 0.12$ |
| VARAN (Ours) | | $\mathbf{3.18 \pm 0.06}$ | $6.47 \pm 0.09$ |
| Last Layer | | $4.50$ | $9.73$ |
| Weighted Sum | Data2Vec Large | $7.44$ | $19.92$ |
| VARAN (Ours) | | $\mathbf{4.06}$ | $\mathbf{9.00}$ |

## 4.3 SPEAKER VERIFICATION

To measure the performance of our method on the Speaker Verification task, we train models on the VoxCeleb1 (Nagrani et al., 2017) dataset. For training efficiency purposes, we randomly cut audio segments of 5 seconds at each training step. For the validation set, we removed 11 speakers from the original training set and created 5,500 pairs from all audio recordings of these speakers. Each pair is labeled to indicate whether the audio belongs to the same speaker or not. For the testing set, we used the original test set, which consists of 20,000 audio pairs. To measure the model performance on the task, we used the equal error rate (EER) and the minimum detection cost function with the target probability $P_{tar}$ computed from the labels. Both $C_{fa}$ and $C_{fr}$ share an equal weight of 1.0. Results are presented in Table 2.

According to all metrics, VARAN shows superior performance. Due to high variance, we also report median values for all metrics. We found that the last layer baseline performs the worst, and these results are consistent with previous work (Yang et al., 2024; Pasad et al., 2023), since we need low-

---

[1] https://github.com/microsoft/UniSpeech

Table 2: Speaker verification task with different layer aggregation methods evaluated on Vox-Celeb1 (Nagrani et al., 2017) dataset. VARAN outperforms other methods in terms of both EER and DCF. We also report median value across 5 seeds. See Section 4.3 for more details.

| Method | EER ($\downarrow$) | DCF ($\downarrow$) |
|---|---|---|
| Last Layer | $2.48 \pm 0.25$ / median: 2.46 | $0.048 \pm 0.001$ / median: 0.049 |
| Weighted Sum | $1.55 \pm 0.09$ / median: 1.60 | $0.030 \pm 0.002$ / median: 0.031 |
| Weighted Hiddens | $5.39 \pm 3.85$ / median: 4.84 | $0.107 \pm 0.008$ / median: 0.049 |
| MHFA | $1.52 \pm 0.10$ / median: 1.50 | $0.030 \pm 0.002$ / median: 0.030 |
| VARAN (Ours) | $\mathbf{1.27 \pm 0.21}$ / **median: 1.18** | $\mathbf{0.025 \pm 0.004}$ / **median: 0.023** |

level features from the first layers for speaker verification. Weighted Sum and MHFA (Peng et al., 2022) have almost similar performance, with MHFA slightly outperforming Weighted Sum.

## 4.4 EMOTION RECOGNITION

For the emotion recognition task, we chose the IEMOCAP (Zadeh et al., 2018) dataset, which consists of approximately 12 hours of data across 5 sessions, with 2 speakers per session. We follow the evaluation setup recommendations described by Antoniou et al. (2023), using audio recordings from 5 classes: angry, neutral, sad, happy, and excited. We combine the last two classes into a single class, resulting in a total of 4 classes. We apply 10-fold cross-validation, leaving one speaker in each session for evaluation and another for testing. For hyperparameter search, we use 2-fold cross-validation, training models on 4 sessions and leaving either a male or female speaker for validation. During hyperparameter search, we measure performance based solely on the results from the validation speaker.

Table 3: Emotion Recognition task with different Layer Aggregation methods evaluated on IEMO-CAP (Zadeh et al., 2018) dataset. Last Layer baseline has on-par performance with our method. See Section 4.4 for more details.

| Method | Backbone | Balanced Accuracy ($\uparrow$) | Accuracy ($\uparrow$) | Macro F1 ($\uparrow$) | Weighted F1 ($\uparrow$) |
|---|---|---|---|---|---|
| Last Layer | | 0.758 | 0.748 | 0.747 | 0.748 |
| Weighted Sum | | 0.749 | 0.734 | 0.743 | 0.732 |
| Weighted Hiddens | WavLM Large | 0.714 | 0.686 | 0.685 | 0.680 |
| MHFA | | 0.756 | 0.745 | 0.741 | 0.752 |
| VARAN (Ours) | | **0.780** | **0.758** | **0.764** | **0.754** |
| Last Layer | | 0.606 | 0.567 | 0.572 | 0.562 |
| Weighted Sum | Data2Vec Large | 0.631 | 0.596 | 0.599 | 0.586 |
| VARAN (Ours) | | **0.684** | **0.645** | **0.653** | **0.645** |

Even though the MHFA layer-aggregation method was initially proposed for the SV task, we found that it generalizes well to the SER task. The results are presented in Table 3. In our results, we observed high variance from fold to fold, so we reported the median result. VARAN and Last Layer perform best. We believe that, unlike in the speaker verification task (see Section 4.3), the model classifies emotion based on high-level features such as phonemes. Previous research by Sun et al. (2021) has shown that incorporating text modality into speech emotion recognition can improve performance. This is further supported by the posterior distribution learned for the emotion recognition task. The best performance was achieved when the prior and posterior distributions gave more weight to the last layers (see Section 5.1). The full results can be found in Appendix F.

The VARAN layer-aggregation method can be easily extended to other speech self-supervised models. To demonstrate this, we report the results obtained on the Data2Vec Baevski et al. (2022) Large model. For the experiments, we used the model weights for the pretrained feature extractor[2]. As shown in Table 3, the VARAN layer-aggregation method significantly outperforms the Weighted Sum and Last Layer methods. Notably, the Last Layer method performs the worst for the Data2Vec model.

## 5 ANALYSIS

### 5.1 PRIOR DISTRIBUTION IMPACT

Firstly, we decided to explore the impact of the prior distribution on performance, as it can be seen as an additional hyperparameter. For that, we compare the second-best layer aggregation method for each task and VARAN trained with the 3 distributions: reversed geometric distribution, uniform distribution and the scaled non-central $\chi^2$ distributions. For each distribution, we used grid-search to find the optimal hyperparameters, including distribution-specific parameters such as probability of success $p$ for the geometric distribution and the non-centrality parameter $\lambda$ for $\chi^2$ the distribution. All models are trained with WavLM Large (Chen et al., 2022) as a backbone model.

Table 4: Emotion Recognition task with different priors for VARAN method evaluated on IEMO-CAP (Zadeh et al., 2018) dataset. Note that best results for emotion recognition task are achived with the non-cetal $\chi^2$ distribution. See Section 5.1 for more details.

| Method | Prior Distribution | Balanced Accuracy ($\uparrow$) | Accuracy ($\uparrow$) | Macro F1 ($\uparrow$) | Weighted F1 ($\uparrow$) |
|---|---|---|---|---|---|
| Last Layer | - | 0.758 | 0.748 | 0.747 | 0.748 |
| VARAN | Uniform | 0.715 | 0.692 | 0.699 | 0.718 |
|  | Reversed geometric | 0.751 | 0.737 | 0.740 | 0.731 |
|  | **Non-central $\chi^2$** | **0.780** | **0.758** | **0.764** | **0.754** |

Table 4 shows the results for the emotion recognition task. We report the median score based on 10-fold cross-validation. The model that achieved the best score used the VARAN layer-aggregation method with a non-central $\chi^2$ distribution as the prior. When using other prior distributions, VARAN's performance decreased significantly, and the last layer aggregation method outperformed the other two VARAN models.

The results for the speech recognition task can be found in Table 5. In contrast to emotion recognition, our experiments indicate that the best suitable prior distribution for the ASR task is a reversed geometric distribution.

The main difference between the non-central $\chi^2$ distribution and the reversed geometric distribution is that in the non-central $\chi^2$ prior distribution's probability mass function starts to slightly decrease after one of the last layers. In contrast, in the reversed geometric distribution, the layer weight increases up to layer $N$.

This finding aligns with the results reported by Yang et al. (2024), who used the i'th layer for model inference. They found that the performance of several speech self-supervised models improved with each subsequent layer, while the results for emotion recognition improved up to a certain layer and then started to decrease. Thus, the chosen prior distribution for the VARAN layer-aggregation method correlates with the layer-model performance, in contrast to the weights learned within the Weighted Sum method (Yang et al., 2024).

### 5.2 REGULARIZATION STRENGTH

The objective function for the VARAN layer-aggregation method combines the task loss and the KL divergence between the chosen prior and the predicted posterior distribution (see Equation 1). An important hyperparameter in our training pipeline is $\beta$, which weights the regularization of the

---

[2]https://huggingface.co/facebook/data2vec-audio-large

Table 5: VARAN performance on Speech Recognition task with different prior distributions. See Section 5.1 for more details.

| Method | Prior Distribution | WER clean ($\downarrow$) | WER other ($\downarrow$) |
|---|---|---|---|
| Weighted Sum | - | 3.28 | **6.34** |
| VARAN | Uniform | 3.40 | 6.99 |
| | Reversed geometric | **3.24** | 6.56 |
| | Non-central $\chi^2$ | 3.36 | 7.12 |

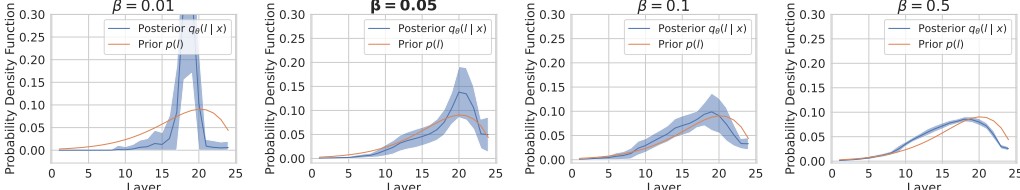

Figure 2: Learned posterior distribution with different regularization term scale $\beta$ for SER task on IEMOCAP dataset. Standard deviation across samples is indicated by filling. $\beta = 0.05$ performs best. If $\beta$ value is too low, distribution collapse into a single mode solution, which leads to low performance. If the regularization scale is too large, it leads to low variance across samples. See Section 5.2 for more details.

posterior distribution. To assess the impact of posterior distribution regularization, we trained 5 models with the same non-central $\chi^2$ distribution but varied $\beta$ values on SER task. Figure 2 shows examples of the learned posterior distributions. The model with $\beta = 0.05$ performed best. Note that VARAN selects a different posterior distribution for each sample, and finding the optimal $\beta$ is crucial for this adaptability. We found that lower $\beta$ values are tend to collapse into a single layer distribution, in contrast with large $\beta$ values posterior distribution collapse to prior with almost zero standard deviation across samples.

### 5.3 TRUNCATING POSTERIOR DISTRIBUTION

Prior distributions we used have non-zero probabilities for every layer. However, we suppose that, in some cases, it is better to zero-out scores from certain classifiers. Therefore, we decided to validate how truncating prior distribution via top-k would affect performance. More formally, in this experiment we use

$$\log \hat{p}(y|x) = \sum_{i \in top_k(q(l|x))} q_\theta(i \mid x) \log p_\theta(y \mid x, l = i)$$

as a prediction of our model on inference.

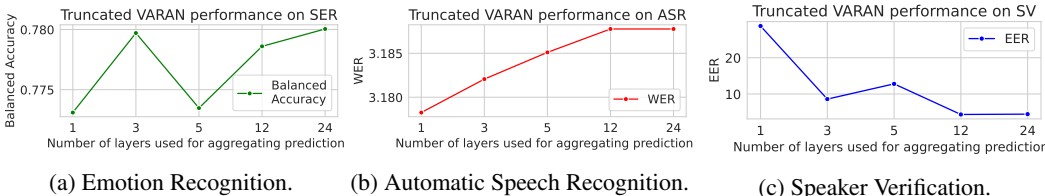

(a) Emotion Recognition.      (b) Automatic Speech Recognition.      (c) Speaker Verification.

Figure 3: VARAN Performance with truncated objective function. See Section 5.3 for more details. SV and ER tasks perform better with no truncation, when features from all layers are utilized, however ASR task needs only a single layer to give the best performance.

Figure 3 shows how selection of only top-k layers by value of posterior distribution from Transformer Encoder affects the performance of our method on inference. Such dependency of performance on the number of layers can be explained by the nature of the task. If the task requires the model to utilize different features (e.g., emotion recognition and speaker verification), it is better to get information from all the layers. If the task requires similar features (e.g., speech recognition), VARAN will perform better utilizing features from a single layer. It can be seen that the best results for emotion recognition and speaker verification tasks are produced when all Transformer layers are utilized. In speech recognition, increasing the number of layers does not lead to improved performance, because this task requires similar features.

## 5.4 SHARED WEIGHTS

For VARAN inference for each layer, we train a separate projector and classifier. In this section, we observe how sharing weights among projectors and classifiers influences VARAN's performance on SER and SV tasks.

Table 6: Emotion Recognition with VARAN, VARAN-ShW with shared projectors' and classifiers' weights and second-best Layer Aggregation method evaluated on IEMOCAP (Zadeh et al., 2018) dataset. Note that according to F1 Weighted Score, VARAN's performance doesn't degrade after sharing weights. For other metrics, the performance drops slightly, but still outperforms the baseline method. See Section 5.4 for more details.

| Method | Balanced Accuracy (↑) | Accuracy (↑) | Macro F1 (↑) | Weighted F1 (↑) |
|---|---|---|---|---|
| Last Layer | 0.758 | 0.748 | 0.747 | 0.748 |
| VARAN-ShW | 0.766 | 0.755 | 0.759 | **0.754** |
| VARAN | **0.780** | **0.758** | **0.764** | **0.754** |

It can be seen that, according to most metrics, VARAN's performance while the weights of classifiers and projectors are shared drops on both SV and SER tasks. However, the performance on emotion recognition tasks is insufficient, and VARAN still outperforms the best of the compared aggregation methods (see Table 6). For SV, the decrease in metrics is more notable, and the performance of VARAN while weights among projectors and classifiers are shared is comparable to MHFA.

Table 7: Speaker verification task with VARAN, VARAN-ShW with shared weights of projectors and classifiers, and second-best MHFA layer aggregation method evaluated on the VoxCeleb1 (Nagrani et al., 2017) dataset. Note that while parameters among projectors and classifiers are shared in the SV task, VARAN's performance drops, and its metrics are almost like on MHFA. See Section 5.4 for more details.

| Method | EER% (↓) | DCF% (↓) |
|---|---|---|
| MHFA | $1.52 \pm 0.10$ / median: 1.50 | $3.00 \pm 0.17$ / median: 2.95 |
| VARAN (Sh-W) | $1.53 \pm 0.18$ / median: 1.44 | $3.01 \pm 0.34$ / median: 2.78 |
| VARAN | $\mathbf{1.27 \pm 0.21}$ / **median: 1.18** | $\mathbf{2.52 \pm 0.42}$ / **median 2.34** |

For speaker verification task, VARAN's performance with shared weights decreases by a noticeable value, but it is still almost identical compared to the second-best method, MHFA, in terms of mean values, and outperforms it in terms of median values (see Table 7).

## 6 RELATED WORK

### 6.1 SPEECH SELF-SUPERVISED MODELS FINE-TUNING TECHNIQUES

To fine-tune an SSL model for a downstream task, one approach is to keep the model parameters frozen and fine-tune only the downstream architecture, commonly used for benchmarking different

upstream models (wen Yang et al., 2021a). While efficient, it doesn't always yield the best performance. To improve performance, task-specific downstream architectures like TDNN and ECAPA-TDNN have been introduced for speaker verification (SV) (Fan et al., 2021). These architectures perform well on raw mel-spectrograms (Desplanques et al., 2020) and even better when combined with SSL models (Novoselov et al., 2022).

Sometimes, fine-tuning only the downstream model is not enough for state-of-the-art performance. For example, fine-tuning Self-Supervised Speech Models for automatic speech recognition (ASR) often involves unfreezing the Transformer Encoder (Chen et al., 2022; wen Yang et al., 2021a). Unfreezing the backbone model's weights also helps achieve strong performance on many downstream tasks using lightweight architectures (Li et al., 2022; Manila et al., 2024).

In this work, we explore enhanced techniques to fine-tune pretrained speech self-supervised models by reweighting Transformer Encoder outputs. The closest related work is by Peng et al. (2022), who proposed a lightweight architecture to enhance the performance of speech self-supervised models on a speaker verification task using a layer aggregation method (see Section 5.2).

## 6.2 LAYER ANALYSIS OF SPEECH SELF-SUPERVISED MODELS

Chen et al. (2022) found that layers at the beginning and middle of the WavLM model had higher magnitudes of learned weights after fine-tuning on speaker-dependent tasks, indicating their importance for speaker verification (SV) and speaker identification (SI). Pasad et al. (2023) used CCA analysis to show that models trained to predict discrete units learn phonetic and word information in intermediate layers, with this information concentrated in higher layers. They also demonstrated that layer-wise phone and word content correlate well with downstream task performance for phoneme recognition (PR) and automatic speech recognition (ASR). Yang et al. (2024) showed that weights learned via weighted sum are not correlated with layers' performance on tasks. Both Yang et al. (2024) and Pasad et al. (2023) found that for some tasks and SSL models, the best individual layer outperforms the learned weighted sum. However, this improvement in single-layer benchmarking is specific to individual SSL models and is not consistent across all models, except for the VC task (Pasad et al., 2023).

So far, there have been few attempts to utilize the observed knowledge about layer-dependent performance. One such attempt is MHFA (Peng et al., 2022) propped on the SV task, where two sets of weights were introduced to separately encode speaker-discriminative information and phonetic units. See the described method in more detail in Section 2.2. Despite these efforts, using a weighted sum for SSL model training remains the most efficient and widely used approach across all tasks and models.

## 6.3 VARIATIONAL INFERENCE AND EARLY EXIT

From one perspective, our method could be seen as VAE (Kingma & Welling, 2014) with layer index $l$ as a latent variable. Therefore, we can successfully adapt some techniques for it, like Higgins et al. (2017). Banino et al. (2021) used a similar approach, but for adaptive computational time (Graves, 2016) inference. This method was also adapted for early exiting during inference (Balagansky & Gavrilov, 2022). While reducing inference time of the models is a promising direction, in our work, we concentrated solely on performance improvement.

## 7 CONCLUSION

In our work, we present a novel method called VARAN for utilizing self-supervised speech models on downstream tasks. Our method is based on variational inference for choosing the weights of the predictions from all backbone layers in a per-sample manner. Through extensive experiments, we show that our method consistently outperforms other methods for tasks where we need to process both low-level features (i.e., speaker verification) and high-level features (i.e., speech or emotion recognition). We also conduct an in-depth analysis of the proposed methods and provide valuable insights. Our method does not depend on its downstream task and can be used for any of the current speech tasks.

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

## A    ELBO

For simplicity, let us denote approximated posterior $q_\theta$ as $q$, downstream task distribution $p_\phi(\cdot|\cdot)$ as $p(\cdot|\cdot)$, prior as $p(\cdot)$ and $\mathbb{E}_{l \sim q_\theta(l|x)}$ as $\mathbb{E}_q$ .

First, let us get a formula for KL-divergence between the approximated distribution and the true posterior:

$$\mathrm{KL}\left[q(l\mid x)\|p(l\mid x,y)\right] = \mathbb{E}_q \log \frac{q(l|x)}{p(l\mid x,y)} = \mathbb{E}_q \log \frac{q(l|x) \cdot p(x,y)}{p(l,y,x)} =$$

$$= \mathbb{E}_q \log \frac{q(l|x) \cdot p(y|x) \cdot p(x)}{p(l,y|x) \cdot p(x)} = \mathbb{E}_q \log \frac{q(l|x) \cdot p(y|x)}{p(l,y|x)} =$$

$$= \mathbb{E}_q \log \frac{q(l|x)}{p(l,y|x)} + \mathbb{E}_q \log p(y|x) = \mathbb{E}_q \log \frac{q(l|x)}{p(l,y|x)} + \log p(y|x).$$

With terms rearranged:

$$\log p(y|x) = \mathbb{E}_q \log \frac{p(l,y|x)}{q(l|x)} + \mathrm{KL}\left(q(l|x))\|p(l|x,y)\right).$$

From this equation, the lower bound for likelihood can be written as:

$$LB = -L(\theta) = \mathbb{E}_q \log \frac{p(l,y|x)}{q(l|x)}$$

Let us further simplify this LB to bring it to a form that can be used to update gradients. First, we need to express our LB through KL-divergence:

$$LB = -L(\theta) = \mathbb{E}_q \log \frac{p(l, y|x)}{q(l|x)} = \mathbb{E}_q \log \frac{p(y|x, l) \cdot p(l)}{q(l|x)} = \mathbb{E}_q \log p(y|x, l) + \mathbb{E}_q \log \frac{p(l)}{q(l|x)}$$

From the definition of KL-divergence, $\text{KL}\left(q(l|x)\|p(l)\right) = \mathbb{E}_q \log \frac{q(l|x)}{p(l)}$, we get that

$\mathbb{E}_q \log \frac{p(l)}{q(l|x)} = -\text{KL}\left(q(l|x)\|p(l)\right)$. Which gives us equation 2 that is the same as equation 1:

$$L(\theta) = -\mathbb{E}_q \log p(y|x, l) + \text{KL}\left(q(l|x)\|p(l)\right) \tag{2}$$

By expanding $\mathbb{E}_q$ and selecting the downstream model, $p_\theta$, we get equation 3:

$$L(\theta) = -\sum_{i=1}^{L} q(l = i|x) \cdot \log p_\theta(y|l = i, x) + \text{KL}\left(q(l|x)\|p(l)\right) \tag{3}$$

## B    PARAMETERS AND PERFORMANCE OF THE METHODS

| Method | Total Parameters | Trainable Parameters | Forward Time (ms) | Overhead |
|--------|------------------|----------------------|-------------------|----------|
| Last Layer | 315 717 319 | 311 510 727 | $26.647 \pm 0.722$ | 0.0% |
| Weighted Sum | 315 717 344 | 311 510 752 | $26.828 \pm 0.515$ | $\sim 0.7\%$ |
| VARAN (Ours) | 323 103 681 | 318 897 089 | $27.992 \pm 0.278$ | $\sim 4.8\%$ |

Table 8: The number of parameters and forward pass time are measured for the RAVDESS (Livingstone & Russo, 2018) Dataset with a batch size of 1. Note that VARAN introduces an overhead of about 5% in the validation forward pass compared to the last layer, which is the fastest layer aggregation method. We observe similar behavior on other datasets and tasks, as the VARAN layer aggregation method is almost identical for all tasks.

## C    PRIOR DISTRIBUTION

In all cases except the ASR task, we use the $\chi^2$ distribution obtained by cropping the distribution with 0.01 and 0.99 quantiles, then rescaling it into layer index range (from 1 to 24) and then discretizing it to a categorical distribution with weights proportional to the PDF of the continuous distribution. We then vary a non-centricity parameter to obtain different prior distributions. Probability functions of these distributions are presented in Figure 4.

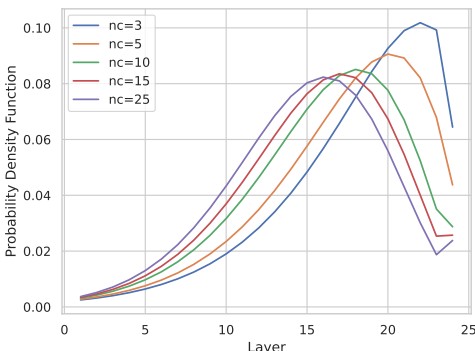

Figure 4: $\chi^2$-based prior distributions obtained by procedures described in Appendix C.

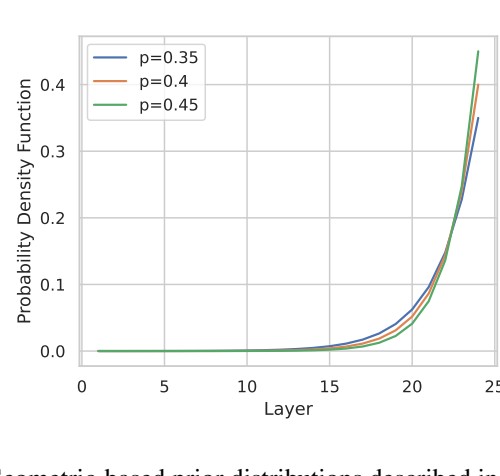

Figure 5: Geometric-based prior distributions described in Appendix C.

For the ASR tasks, we use the reversed geometric distribution with various $p$ parameters. We hypothesized that, we only need high-level features for the ASR task, and therefore it is better for prior distribution to have a mode on the last layer. Probability density functions of geometric-based distributions can be found in Figure 5.

## D   VARAN EXPECTED VALIDATION PERFORMANCE

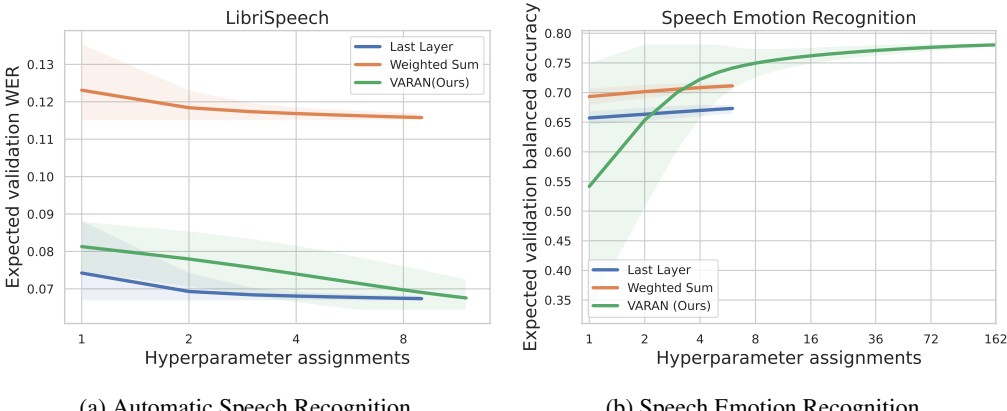

(a) Automatic Speech Recognition.

(b) Speech Emotion Recognition.

Figure 6: VARAN's Expected Validation Performance on the ASR and SER tasks shows that it performs on par with the Last Layer baseline in ASR. However, given a sufficient budget, VARAN begins to outperform this baseline after approximately 10 hyperparameter assignments. In the SER task, VARAN surpasses the Last Layer baseline after just 2 hyperparameter assignments and outperforms the Weighted Sum baseline after 3 assignments.

VARAN requires an additional hyperparameter search, and thus direct comparison with baselines after grid search is not sufficient to demonstrate if our method performs best. We therefore utilize the Expected Validation Performance (EVP) (Dodge et al., 2019) to validate our claims. EVP uses validation results to predict which method would perform best for every grid-search budget. We report the results in Figure 6.

## E   HYPERPARAMETERS GRIDS

Hyperparameters selected for all tasks and methods discussed in experiments, as well as the best hyperparameters, can be found in Tables 9, 10 and 11. See Section 4 for the experiments results.

We also provide the hyperparameter grids used for search, see Tables 12, 13 and 14.

Table 9: Hyperparameters for ASR for different methods. Supplementary for Section 4.2.

| Parameter | VARAN | Last Layer | Weighted Sum | Weighted Hiddens |
|---|---|---|---|---|
| num_heads_mhsa_prior_dist_pred | 16 | - | - | 16 |
| downstream_head | Linear | Linear | Linear | Linear |
| projector_dim | 128 | 128 | 128 | 128 |
| prior_distribution | geometric | - | - | - |
| p_geometric_distribution | 0.45 | - | - | - |
| reversed_prior_distribution | True | - | - | - |
| epoch | 25 | 25 | 25 | 25 |
| scheduler | cosine_annealing | cosine_annealing | cosine_annealing | cosine_annealing |
| effective_batch_size | 16 | 16 | 16 | 16 |
| lr | 1.00E-05 | 1.00E-05 | 1.00E-05 | 1.00E-05 |
| clip_grad_value | 1.0 | 1.0 | 1.0 | 1.0 |
| kl_beta | 0.01 | - | - | - |
| optimizer | AdamW | AdamW | AdamW | AdamW |
| beta1 | 0.9 | 0.9 | 0.9 | 0.9 |
| beta2 | 0.999 | 0.999 | 0.999 | 0.999 |
| weight_decay | 0.1 | 0.1 | 0.1 | 0.1 |
| freeze_cnn | True | True | True | True |
| freeze_upstream | False | False | False | False |
| task_loss | CTC | CTC | CTC | CTC |
| dictionary_len | 32 | 32 | 32 | 32 |
| unit_lm | False | False | False | False |
| decoding | Beam Search | Beam Search | Beam Search | Beam Search |
| beam_size | 5 | 5 | 5 | 5 |
| beam_threshold | 20 | 20 | 20 | 20 |

Table 10: Hyperparameters for SER for different methods. Supplementary for Section 4.4.

| Parameter | VARAN | Last Layer | Weighted Sum | Weighted Hiddens | MHFA |
|---|---|---|---|---|---|
| num_heads_mhsa_prior_dist_pred | 16 | - | - | 16 | - |
| downstream_head | MLP | MLP | MLP | MLP | Linear |
| projector_dim | 128 | 256 | 128 | 128 | - |
| prior_distribution | $\chi^2$ | - | - | - | - |
| chi2_degrees_of_freedom | 2 | - | - | - | - |
| chi2_lambda_non_centrality_parameter | 10 | - | - | - | - |
| max_epoch | 15 | 15 | 15 | 15 | 15 |
| scheduler | cosine_annealing | cosine_annealing | cosine_annealing | cosine_annealing | cosine_annealing |
| effective_batch_size | 32 | 32 | 32 | 128 | 128 |
| lr | 1.00E-04 | 1.00E-05 | 1.00E-05 | 1.00E-04 | - |
| clip_grad_value | 1.0 | 1.0 | 1.0 | 1.0 | 1.0 |
| kl_beta | 0.05 | - | - | - | - |
| optimizer | AdamW | AdamW | AdamW | AdamW | AdamW |
| beta1 | 0.9 | 0.9 | 0.9 | 0.9 | 0.9 |
| beta2 | 0.999 | 0.999 | 0.999 | 0.999 | 0.999 |
| weight_decay | 0.01 | 0.01 | 0.01 | 0.01 | - |
| freeze_cnn | True | True | True | True | True |
| freeze_upstream | False | False | False | False | False |
| task_loss | weighted_crossentropy | weighted_crossentropy | weighted_crossentropy | weighted_crossentropy | weighted_crossentropy |
| llrd_factor | - | - | - | - | 1.2 |
| lr_transformer | - | - | - | - | 2.00E-05 |
| lr_mhfa | - | - | - | - | 1.00E-03 |
| weight_finetuning_reg | - | - | - | - | 1.00E-03 |
| head_nb | - | - | - | - | 64 |
| ce_weights | ang 1.25, neu 0.85, hap 0.81, sad 1.28 | ang 1.25, neu 0.85, hap 0.81, sad 1.28 | ang 1.25, neu 0.85, hap 0.81, sad 1.28 | ang 1.25, neu 0.85, hap 0.81, sad 1.28 | ang 1.25, neu 0.85, hap 0.81, sad 1.28 |

## F ADDITIONAL RESULTS

As mentioned in Section 4.4, we found that standard deviation across different sets is high (see Table 15).

As an additional experiment for the speech emotion recognition task, we decided to observe how posterior distribution is dependent on the emotion we are trying to classify. Results are presented in Figure 7.

Table 11: Hyperparameters for SV for different methods. Supplementary for Section 4.3.

| Parameter | VARAN | Last Layer | Weighted Sum | Weighted Hiddens | MHFA |
|---|---|---|---|---|---|
| num_heads_mhsa_prior_dist_pred | 16 | - | 16 | - | - |
| downstream_head | Linear | Linear | Linear | Linear | Linear |
| projector_dim | 256 | 128 | 256 | 256 | - |
| prior_distribution | $\chi^2$ | - | - | - | - |
| chi2_degrees_of_freedom | 2 | - | - | - | - |
| chi2_lambda_non_centrality_parameter | 10 | - | - | - | - |
| max_epoch | 15 | 15 | 15 | 15 | 10 |
| scheduler | cosine_annealing | cosine_annealing | cosine_annealing | cosine_annealing | cosine_annealing |
| effective_batch_size | 32 | 64 | 64 | 64 | 32 |
| lr | 1.00E-05 | 1.00E-04 | 1.00E-05 | 5.00E-05 | - |
| clip_grad_value | - | - | - | - | - |
| kl_beta | 0.5 | - | - | - | - |
| optimizer | AdamW | AdamW | AdamW | AdamW | AdamW |
| beta1 | 0.7 | 0.7 | 0.7 | 0.7 | 0.9 |
| beta2 | 0.999 | 0.999 | 0.999 | 0.999 | 0.999 |
| weight_decay | 0.01 | 0.01 | 0.01 | 0.01 | - |
| freeze_cnn | True | True | True | True | True |
| freeze_upstream | False | False | False | False | False |
| task_loss | aamsoftmax | aamsoftmax | aamsoftmax | aamsoftmax | aamsoftmax |
| aamsoftmax_scale | 0.2 | 0.2 | 0.2 | 0.2 | 0.2 |
| aamsoftmax_margin | 30 | 30 | 30 | 30 | 30 |
| llrd_factor | - | - | - | - | 1 |
| lr_transformer | - | - | - | - | 8.00E-05 |
| lr_mhfa | - | - | - | - | 1.00E-03 |
| weight_finetuning_reg | - | - | - | - | 8.00E-04 |
| head_nb | - | - | - | - | 64 |
| train_audio_crop | 5 seconds random crop | 5 seconds random crop | 5 seconds random crop | 5 seconds random crop | 5 seconds random crop |

Table 12: Grid-search parameters for Speech Emotion Recognition Task

| Parameter | Values |
|---|---|
| lr | [1.00E-04, 1.00E-05, 3.00E-04] |
| effective_batch_size | [32, 64, 128] |
| $\beta$ | [0.1, 0.05, 0.01] |
| $\chi^2$ distribution $\lambda$ | [5, 10, 15] |
| projector_dim | [32, 128, 256] |
| llrd_factor | [0.8 1. 1.2] |
| lr_transformer | [1.00E-04, 2.00E-05] |
| lr_mhfa | [1.00E-03, 5.00E-04] |
| head_nb | [16, 64] |

Table 13: Grid-search parameters for Automatic Speech Recognition Task

| Parameter | Values |
|---|---|
| lr | [1.00E-04, 1.00E-05] |
| effective_batch_size | [16, 32, 64] |
| $\beta$ | [0.3, 0.1, 0.05, 0.01] |
| prior_distribution | [geometric, uniform, $\chi^2$] |
| $\chi^2$ distribution $\lambda$ | [1, 5, 10] |
| p_geometric_distribution | [0.3, 0.35, 0.4, 0.45] |
| projector_dim | [32, 128, 256] |

Table 14: Grid-search parameters for Speaker Verification Task

| Parameter | Values |
|---|---|
| lr | [1.00E-04, 1.00E-05, 5.00E-05] |
| effective_batch_size | [32, 64, 128] |
| $\beta$ | [0.1, 0.05, 0.01] |
| $\chi^2$ distribution $\lambda$ | [5, 10, 15] |
| projector_dim | [32, 128, 256] |
| llrd_factor | [1. 1.2] |
| lr_transformer | [8.00E-05, 2.00E-05] |
| lr_mhfa | [1.00E-03, 5.00E-04] |
| weight_finetuning_reg | [8.00E-04, 1.00E-04] |

Table 15: Emotion Recognition task with different Layer Aggregation methods. Mean values are shown for corresponding metrics. Evaluated on IEMOCAP (Zadeh et al., 2018) dataset, WavLM backbone with linear head. See Section 5.1 for more details.

| Method | Backbobe | Balanced Accuracy ($\uparrow$) | Accuracy ($\uparrow$) | Macro F1 ($\uparrow$) | Weighted F1 ($\uparrow$) |
|---|---|---|---|---|---|
| Last Layer | | $0.766 \pm 0.027$ | $\mathbf{0.750 \pm 0.039}$ | $\mathbf{0.753 \pm 0.038}$ | $\mathbf{0.748 \pm 0.041}$ |
| Weighted Sum | | $0.745 \pm 0.035$ | $0.734 \pm 0.033$ | $0.738 \pm 0.036$ | $0.733 \pm 0.034$ |
| Weighted Hiddens | WavLM Large | $0.712 \pm 0.038$ | $0.691 \pm 0.045$ | $0.693 \pm 0.043$ | $0.687 \pm 0.048$ |
| MHFA | | $0.756 \pm 0.036$ | $0.742 \pm 0.044$ | $0.74 \pm 0.046$ | $\mathbf{0.748 \pm 0.042}$ |
| VARAN (Ours) | | $\mathbf{0.772 \pm 0.029}$ | $0.748 \pm 0.033$ | $0.750 \pm 0.033$ | $0.745 \pm 0.034$ |
| Last Layer | | $0.592 \pm 0.053$ | $0.555 \pm 0.07$ | $0.558 \pm 0.068$ | $0.543 \pm 0.074$ |
| Weighted Sum | Data2Vec Large | $0.639 \pm 0.037$ | $0.599 \pm 0.042$ | $0.597 \pm 0.041$ | $0.585 \pm 0.048$ |
| VARAN (Ours) | | $\mathbf{0.680 \pm 0.043}$ | $\mathbf{0.650 \pm 0.045}$ | $\mathbf{0.647 \pm 0.052}$ | $\mathbf{0.639 \pm 0.054}$ |

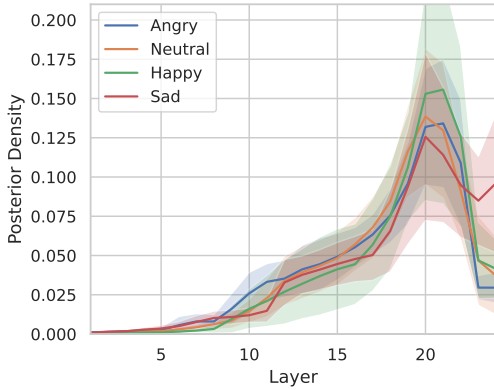

Figure 7: Posterior distribution of different emotions on the validation dataset. We observe that the emotion "angry" tends to be classified in the earlier stages. Layers 20 and 21 have more probabilities in the case of emotion "happy". Emotion "sad" tends to be classified on the last layers. See Appendix F for more details.

To demonstrate VARAN's generalizability across languages, we provide experiments on a subset of the Common Voice (Ardila et al., 2020) dataset, version 17[3] in the Russian language, which consists of approximately 27 hours. The results shown in Table 16 indicate that VARAN significantly outperforms other methods. Note that the Russian language was not present in the WavLM pretraining corpus and can be considered an out-of-domain distribution. For the prior distribution for the VARAN layer aggregation method, we chose a reversed geometric distribution, as for the English language ( 4.2), with a probability of success $p = 0.55$.

To demonstrate the generalizability across datasets we provide an experiments for SER task on RAVDESS corpus (Livingstone & Russo, 2018). We followed the evaluation setup described in (Koolagudi & Rao, 2012). We merged the neutral and calm emotions, resulting in 7 emotions, and used the first 20 actors for training, actors 20-22 for best hyperparameter search on validation and actors 22-24 for test. Results are present in Table **??** . Note that VARAN outperforms other methods across both upstream models. As a prior distribution, for both data2vec and WavLM models we used reversed non-central $\chi^2$ distribution, the same is used in IEMOCAP dataset in Section 4.4 with degrees of freedom 2 and non cenntrality parameter $\lambda$=15.

From these results, we can conclude that the VARAN method and the choice of prior distribution are generalized for other datasets and languages.

Table 16: WER for different layer aggregation methods trained on Speech Recognition task. All models are trained on Russian part of Common Voice training corpus and used WavLM Chen et al. (2022) as backbone.

| Method | WER ($\downarrow$) |
|---|---|
| Last Layer | 51.21 |
| Weighted Sum | 39.82 |
| VARAN (Ours) | **31.78** |

Table 17: Emotion Recognition task with different Layer Aggregation methods evaluated on RAVDESS (Livingstone & Russo, 2018) dataset.

| Method | Backbone | Balanced Accuracy ($\uparrow$) | Accuracy ($\uparrow$) | Macro F1 ($\uparrow$) | Weighted F1 ($\uparrow$) |
|---|---|---|---|---|---|
| Last Layer | | 42.86 | 45.83 | 39.32 | 41.53 |
| Weighted Sum | WavLM Large | 71.42 | 73.33 | 70.26 | 70.82 |
| VARAN (Ours) | | **77.98** | **79.17** | **77.41** | **77.93** |
| Last Layer | | 36.01 | 39.17 | 30.91 | 34.19 |
| Weighted Sum | Data2Vec Large | 58.33 | 60.00 | 54.10 | 55.52 |
| VARAN (Ours) | | **69.94** | **71.33** | **68.80** | **69.77** |

---

[3]https://huggingface.co/datasets/mozilla-foundation/common_voice_17_0

