# OpenReview forum: "Variational Inference for Self-Supervised Speech Models Fine-tuning on Downstream Tasks"
_ICLR.cc/2025/Conference — Submitted to ICLR 2025_

### Official Review · Reviewer_WFfX · 2024-11-01

**Soundness:** 3
**Presentation:** 2
**Contribution:** 3
**Rating:** 5
**Confidence:** 4

**Summary:**

This paper proposes an layer aggregation approach inspired by variational inference. It adds a classifier after each layer along with a final attention layer, and then minimize the variational lower-bound loss to find the best importance distribution for the outputs from each layer. To validate the approach, the authors conducted a series of experiments on SUPERB benchmark, including ASR, speaker verification, and emotion recognition.

**Strengths:**

* The idea of using variational to generate the impact distribution from all the layer outputs seem to be new and interesting.
* The authors conducted extensive experiments on 3 speech related tasks, along with a set of ablation studies with insightful analysis.

**Weaknesses:**

* For ASR evaluations, the WER improvement is too limited. Usually the 2nd digit after the decimal point differences are mostly noise. In this case, it would be helpful to run experiments on other testsets (e.g., common voice, TED-LIUM) to verify if we are actually seeing performance gains.
* The proposed approach requires running additional multi-headed attention during serving time. How many more parameters are in this layer? What’s the latency impact?
* The presentation needs improvement. Please see my questions below. In addition, the title and abstract sound a bit confusing. The proposed approach is on layer aggregation, but neither the title or the abstract mentioned this.

**Questions:**

General question: Is the proposed layer aggregation approach applicable to SSL model? Feels like they can work for models trained from scratch?

Section 2.2: How does it enforce V to learn speaker-discriminative information only, while K has phonetic information?

Section 2.2: The authors mentioned that MHFA requires additional requires searching additional hyperparameters. However, in section 5.2, the regularization weight also needs to be tuned. This doesn’t seem to be a fair comparison to prior studies.

Section 3.1: “Choosing the right layer or combination of layers is challenging, as the training data does not have the optimal layer distribution.” Please explain more on this. Does it mean the empirically learned weights from training data (is it from pretraining or finetuning?) are not exactly the same as their real distribution?

Section 3: Please explain how exactly the proposed method works during inference time. Is it using a weighted average of pooled hidden as the input to the final softmax layer?

Section 4:Please explain the baselines. How are the “weighted” exactly computed?

---

> ### Author Response · Authors · 2024-11-17
>
> Dear Reviewer,
>
> Thank you for your time and for providing a detailed review. We appreciate you recognize the strengths of our work, including its interesting insights, the novel use of variational methods for impact distribution from layer outputs, and the extensive experiments and ablation studies on three speech-related tasks.
>
> Regarding the weaknesses you pointed out and your questions:
>
> 1. **Is the proposed layer aggregation approach applicable to SSL model? Feels like they can work for models trained from scratch?**
>     - We aimed to propose methods for pre-trained speech self-supervised models, as questions about layer importance for different downstream tasks have been raised (see Section 6.2). However, in general, the method can be easily adapted to any transformer-based model, whether pre-trained or trained from scratch, for different tasks. Let us note, that training model from scratch is not a common as pre-trained SSL-models has superior performance.
> 2. **Section 2.2: How does it enforce V to learn speaker-discriminative information only, while K has phonetic information?**
>     - This is an assumption authors do in [1], please refer to Section 3.2 and 4.2.1 in their paper for more details.
>
>    [1] Junyi Peng, Oldrich Plchot, Themos Stafylakis, Ladislav Mosner, Luk´ as Burget, and Jan Cernocky. An attention-based backend
>    allowing efficient fine-tuning of transformer models for speaker verification. In IEEE Spoken Language Technology Workshop, SLT
>    2022, Doha, Qatar, January 9-12, 2023, pp. 555–562. IEEE, 2022. doi: 10.1109/SLT54892.2023.10022775.
> 3. **The authors mentioned that MHFA requires additional requires searching additional hyper-parameters. However, in section 5.2, the regularization weight also needs to be tuned. This doesn’t seem to be a fair comparison to prior studies.**
>     - While both VARAN and MHFA require additional hyper-parameters to be searched, which can be seen as a limitation, VARAN outperforms MHFA on SV and the extended version of MHFA for the SER task (see Tables 2 and 4). To ensure a fair comparison, we also searched for the optimal hyper-parameters for the MHFA layer-aggregation method. Please refer to Appendix D (Tables 11 and 13, where llrd factor, lr transformer, lr mhfa, and weight finetuning reg are all hyper-parameters are MHFA-specific).
> 4. **“Choosing the right layer or combination of layers is challenging, as the training data does not have the optimal layer distribution.” Please explain more on this. Does it mean the empirically learned weights from training data (is it from pretraining or finetuning?) are not exactly the same as their real distribution?**
>
>     - We get our weights through approximating posterior distribution $p(z|x)$ through approximation $q_\theta(z|x)$ learned via variational inference, like in [1]. This weights learned on the fine-tuning stage. Procedure do not require special pre-training. We like to think about the method in a similar to VAE way. VAE aim to train a good representation $z$ from data $x$ ($z$ is not given in the training data), so that we can reconstruct original data from this representation. In our work we aim to “reconstruct” target variable y from learned layer distribution $q(z|x)$ and $x$ itself.
>
>     [1] beta-VAE: Learning Basic Visual Concepts with a Constrained Variational Framework. Irina Higgins and Loic Matthey and Arka Pal and Christopher Burgess and Xavier Glorot and Matthew Botvinick and Shakir Mohamed and Alexander Lerchner. International Conference on Learning Representations 2017.

---

> > ### Author Response · Authors · 2024-11-17
> >
> > 5. **Please explain how exactly the proposed method works during inference time. Is it using a weighted average of pooled hidden as the input to the final softmax layer?**
> >     - During inference, the waveform is passed through the upstream SSL model (e.g., WavLM), which consists of a CNN feature extractor and a Transformer Encoder. All outputs from the Transformer Encoder are collected and passed to the Posterior Distribution Predictor (MHSA block, Figure 1). Formally, this is a list with a size equal to the number of layers, where each element is a tensor with dimensions [batch size, sequence length, hidden dimension]. We then apply mean pooling along the sequence length and concatenate the results into a single tensor, resulting in dimensions [batch size, number of layers, hidden dimension]. This tensor serves as input to the standard MHSA mechanism, functioning as a feature mixer between layers. Finally, we apply a linear layer to the last dimension to obtain a tensor with dimensions [batch size, number of layers, 1], and apply softmax along the number of layers dimension to produce an individual posterior distribution for each sample in the batch. The outputs from different layers of the Transformer Encoder are then summed with the predicted weights, resulting in a tensor of [batch size, sequence length, hidden dimension]. For all tasks except ASR, we then apply mean pooling across the sequence length and obtain final predictions by applying a linear projection from hidden dimension to output size (e.g., number of classes for SER).
> > 6. **Please explain the baselines. How are the “weighted” exactly computed?**
> >     - For the baseline layer-aggregation method, weighted_sum, given weights of shape [n_layers] and an SSL model output of shape [n_layers, batch size, sequence length, hidden dimension], we multiply the weights with the output and sum along the first dimension, resulting in an output of shape [batch size, sequence length, hidden dimension]. n_layers is the number of Transformer Encoder layers (e.g., 24 in WavLM Large).
> > 7. **For ASR evaluations, the WER improvement is too limited. Usually the 2nd digit after the decimal point differences are mostly noise. In this case, it would be helpful to run experiments on other testsets (e.g., common voice, TED-LIUM) to verify if we are actually seeing performance gains.**
> >     - We would like to specify that the results were obtained by training models with several seeds and averaging them, so we would argue that the difference is mostly noise. Moreover, we test the proposed layer aggregation method on three different tasks, each using a separate dataset. This approach is similar to what was used in the WavLM [1] and wav2vec 2.0 [2] papers, where they also demonstrated their models' broad applicability by testing on diverse tasks using single dataset.
> >
> >    [1] Sanyuan Chen, Chengyi Wang, Zhengyang Chen, Yu Wu, Shujie Liu, Zhuo Chen, Jinyu Li, Naoyuki Kanda, Takuya Yoshioka,
> >    Xiong Xiao, et al. Wavlm: Large-scale self-supervised pre-training for full stack speech processing. IEEE Journal of Selected
> >    Topics in Signal Processing, 16(6):1505–1518, 2022.
> >
> >    [2] Alexei Baevski, Yuhao Zhou, Abdelrahman Mohamed, and Michael Auli. wav2vec 2.0: A framework for self-supervised
> >    learning of speech representations. In H. Larochelle, M. Ranzato, R. Hadsell, M.F. Balcan, and H. Lin (eds.), Advances in Neural
> >    In- formation Processing Systems, volume 33, pp. 12449–12460. Curran Associates, Inc., 2020.
> >
> > We have now answered all the questions we could without conducting additional experiments. During the rebuttal stage, we will strive to expand the experiments section as much as possible, incorporating your comments about method evaluation. We will also update you on the number of additional parameters and the latency of the VARAN layer-aggregation method.
> >
> > Thank you once again for your valuable feedback.

---

> ### Comment · Reviewer_WFfX · 2024-11-25
> **Thank you for the responses!**
>
> I would appreciate all the explanations from the authors! They do address some of my questions. However, there are still a couple questions remaining.
>
> 1. From the discussions of question #3, it seems that the proposed approach still needs hyper-params tuning, so this is not an actual benefit compared to prior work. I would suggest to remove this argument from the manuscript to make it a fair comparison to prior work.
>
> 2. There is no experimental results to address question #7 (although it is working in progress), so I would still challenge on the improvements. This is my biggest concern, and I'm happy to adjust my ratings if this question is addressed.

---

> > ### Author Response · Authors · 2024-11-26
> >
> > Thank you for your reply, we would like to inform you that we have uploaded a revised version of the manuscript, which we believe provides additional justification for some of your concerns.
> >
> > > There is no experimental results to address question #7: “For ASR evaluations, the WER improvement is too limited. Usually the 2nd digit after the decimal point differences are mostly noise. In this case, it would be helpful to run experiments on other testsets (e.g., common voice, TED-LIUM) to verify if we are actually seeing performance gains.”
> > >
> >
> > We have updated our work to include several additional results for the ASR task:
> >
> > - We added the Data2Vec Large backbone to our LibriSpeech experiment, and VARAN consistently outperforms other methods, see Table 1 for more details.
> > - We also added an experiment with the subset of Common Voice dataset in Russian language, as it was not included in the WavLM pre-training dataset and can be seen as out of domain distribution (See Table 16).
> >
> > > The proposed approach requires running additional multi-headed attention during serving time. How many more parameters are in this layer? What’s the latency impact?
> > >
> >
> > We included the forward pass time and the number of trainable parameters in Table 8.
> >
> > We highlighted all changes in red for your convenience. If you have any questions, we will be happy to answer them.

---

> ### Author Response · Authors · 2024-11-28
>
> Dear Reviewer "WFfX",
>
> We would like to remind you that we have updated our paper since your last review, and we believe the experiments we have provided address your concerns. In light of these revisions, we kindly ask you to reconsider your score. If you have any further questions, we would be happy to address them.

---

> > ### Author Response · Authors · 2024-12-02
> >
> > Dear Reviewer "WFfX",
> >
> > We would like to remind you that we have updated our paper since your last review, and we believe the experiments we have provided address your concerns. In light of these revisions, we kindly ask you to reconsider your score. If you have any further questions, we would be happy to address them.

---

### Official Review · Reviewer_32Z2 · 2024-11-04

**Soundness:** 3
**Presentation:** 3
**Contribution:** 3
**Rating:** 5
**Confidence:** 3

**Summary:**

This study proposes a novel layer aggregation approach using variational inference. It demonstrates that this method outperforms traditional aggregation strategies across various downstream tasks. By allowing the adjustment of layer importance through a prior distribution, this paper shows that models can be better adapted to task-specific situations.

**Strengths:**

1. This paper introduces a new layer aggregation method that differs from Multi-head Factorized Attentive Pooling layer aggregation by adjusting the prior distribution to control layer output weight.
2. The experiments and analyses demonstrate that the actual posterior distribution follows the prior distribution, and shows good performance on various downstream tasks.

**Weaknesses:**

1. The proposed method (VARAN) involves setting a prior distribution for knowledge distillation (KD) learning, which presents the limitation that one must already know the optimal layer weight distribution for each downstream task.
2. In the ASR experiments, the evaluation was conducted using LibriSpeech data, the domain employed in the pre-training. This method does not align with the goal of assessing generalizability, suggesting a need to test with other speech datasets.
3. This paper conducted experiments on automatic speech recognition, speaker verification, and emotion recognition. Additional datasets are necessary to demonstrate more generalized performance compared to existing methods.

**Questions:**

1. How was the prior distribution chosen, and what were the criteria for selecting the non-central Chi-squared distribution?
2. Why were the WavLM and Data2vec chosen among various self-supervised models, and why did you not experiment with other models?
3. In Figure 2, experiments were conducted to set various beta values for the non-central Chi-squared distribution. Is the optimal beta value found simply by grid search, or if another search method was used?

---

> ### Author Response · Authors · 2024-11-17
>
> Dear Reviewer,
>
> Thank you for your time and for providing a detailed review. We appreciate your recognition of VARAN as a novel layer aggregation method that adjusts the prior distribution to control layer output weight, and your acknowledgment of our experiments showing alignment between the actual posterior and prior distributions, along with strong performance on various downstream tasks.
>
> Regarding the weaknesses you pointed out and your questions:
>
> 1. **How was the prior distribution chosen, and what were the criteria for selecting the non-central Chi-squared distribution?**
>     - Initially, we suggest that VARAN can be used with different downstream tasks and backbone models. Therefore, we designed a process that does not require any prior knowledge of the task, aiming to simplify it by allowing a grid search to find the optimal statistical distribution parameters. The non-central Chi-squared distribution is a good choice because, by adjusting its parameters and the ability to reverse the result distribution, we can construct priors with modes corresponding to different layers.
> 2. **In Figure 2, experiments were conducted to set various beta values for the non-central Chi-squared distribution. Is he optimal beta value found simply by grid search, or if another search method was used?**
>     - Yes, the optimal beta is found simply through grid search. Our results show that $\beta = 0.5$ is a strong coefficient for posterior distribution regularization, making it almost identical to the prior, which causes suboptimal performance. Conversely, setting $\beta < 0.01$ results in too little regularization, also leading to suboptimal performance. Therefore, the suggested values for grid search are between 0.01 and 0.5.
> 3. **In the ASR experiments, the evaluation was conducted using LibriSpeech data, the domain employed in the pre-training. This method does not align with the goal of assessing generalizability, suggesting a need to test with other speech datasets.**
>     - Although audiobooks from the LibriSpeech domain were used during the pre-training stage, we do not believe this impacts the experiments. The advantage of having domain-specific data would equally benefit all layer-aggregation methods, and similarly, all methods might be affected by its absence, but the performance ranking would remain unchanged.
> 4. **This paper conducted experiments on automatic speech recognition, speaker verifcation, and emotion recognition. Additional datasets are necessary to demonstrate more generalized performance compared to existing methods.**
>     - In our paper, we test the proposed layer aggregation method on three different tasks, each using a separate dataset. We believe these experiments are enough to support our claims about generalization to other datasets, as they show the method's effectiveness across various tasks. This approach is similar to what was used in the WavLM [1] and wav2vec 2.0 [2] papers, where they also demonstrated their models' broad applicability by testing on diverse tasks using single dataset.
>
>     [1] Sanyuan Chen, Chengyi Wang, Zhengyang Chen, Yu Wu, Shujie Liu, Zhuo Chen, Jinyu Li, Naoyuki Kanda, Takuya Yoshioka, Xiong Xiao, et al. Wavlm: Large-scale self-supervised pre-training for full stack speech processing. IEEE Journal of Selected Topics in Signal Processing, 16(6):1505–1518, 2022.
>
>     [2] Alexei Baevski, Yuhao Zhou, Abdelrahman Mohamed, and Michael Auli. wav2vec 2.0:
>     A framework for self-supervised learning of speech representations. In H. Larochelle,
>     M. Ranzato, R. Hadsell, M.F. Balcan, and H. Lin (eds.), Advances in Neural In-
>     formation Processing Systems, volume 33, pp. 12449–12460. Curran Associates, Inc.,
>     2020.
>
> 5. **Why were the WavLM and Data2vec chosen among various self-supervised models, and why did you not experiment with other models?**
>     - WavLM and data2vec were chosen from among other speech self-supervised models because they are the newest, widely used, and have achieved the best performance. Architecturally, all speech self-supervised models are quite similar, so other models like wav2vec 2.0 and HuBERT could easily be extended to use the VARAN layer-aggregation method. However, we did not experiment with these other models due to computational constraints.
>
> We have now answered all the questions we could without conducting additional experiments. During the rebuttal stage, we will strive to expand the experiments section as much as possible, incorporating your comments about method evaluation.
>
> Thank you once again for your valuable feedback.

---

> > ### Comment · Reviewer_32Z2 · 2024-11-24
> >
> > The proposed method (VARAN) demonstrates sufficient contribution by enabling controllable layer aggregation through prior distribution. Additionally, the resource constraints cited for conducting less extensive experiments are considered valid. However, the choice of the on-central Chi-squared distribution and the use of grid search to determine the beta value highlight a need for more experimental justification and analysis of the proposed approach. Therefore, the initial score has been maintained due to these limitations.

---

> > > ### Author Response · Authors · 2024-11-26
> > >
> > > Thank you for your reply, we would like to inform you that we have uploaded a revised version of the manuscript, which we believe provides additional justification for some of your concerns.
> > >
> > > > In the ASR experiments, the evaluation was conducted using LibriSpeech data, the domain employed in the pre-training. This method does not align with the goal of assessing generalizability, suggesting a need to test with other speech datasets.
> > > >
> > >
> > > Please note that we added the result for a subset of Common Voice dataset in Russian language. Common Voice dataset and especially Russian speech was not used during WavLM pre-training and can be considered a dataset from out-of-domain distribution. See Table 16 for more details.
> > >
> > > > This paper conducted experiments on automatic speech recognition, speaker verifcation, and emotion recognition. Additional datasets are necessary to demonstrate more generalized performance compared to existing methods.
> > > >
> > >
> > > Despite Common Voice dataset for ASR task we added RAVDESS dataset for SER, the results for both WavLM and Data2Vec models are present Table 17.
> > >
> > > > The choice of the on-central Chi-squared distribution and the use of grid search to determine the beta value highlight a need for more experimental justification and analysis of the proposed approach.
> > > >
> > >
> > > To address concerns regarding the additional hyperparameters and the consequent increase in grid search size, we employ Expected Validation Performance (EVP) [1]. The results are presented in Appendix D (see Figure 6), and we believe this supports our claims and addresses the reviewers' concerns.
> > >
> > > We highlighted all changes in red for your convenience. If you have any questions, we will be happy to answer them.
> > >
> > > [1] Show Your Work: Improved Reporting of Experimental Results. https://arxiv.org/pdf/1909.03004

---

> ### Author Response · Authors · 2024-11-28
>
> Dear Reviewer "32Z2",
>
> We would like to remind you that we have updated our paper since your last review, and we believe the experiments we have provided address your concerns. In light of these revisions, we kindly ask you to reconsider your score. If you have any further questions, we would be happy to address them.

---

> > ### Author Response · Authors · 2024-12-02
> >
> > Dear Reviewer "32Z2",
> >
> > We would like to remind you that we have updated our paper since your last review, and we believe the experiments we have provided address your concerns. In light of these revisions, we kindly ask you to reconsider your score. If you have any further questions, we would be happy to address them.

---

### Official Review · Reviewer_YerL · 2024-11-04

**Soundness:** 3
**Presentation:** 3
**Contribution:** 2
**Rating:** 3
**Confidence:** 5

**Summary:**

This paper introduces VARAN, which demonstrates that different layers encode varying levels of information for different tasks. The authors argue that simple weighted summation is insufficient to capture sample variance. To address this challenge, VARAN automatically learns layer posterior distributions and performs tasks (ASR, SR, and ER) on each layer separately. The aggregated results show improved performance compared to using only the last layer.

**Strengths:**

VARAN adopts variational inference to semi-automatically extract layer-wise information, providing novel insights.


The paper is written in a clear and concise manner, making it easy to follow both the ideas and experiments.


The experimental results support the authors' claims, demonstrating that VARAN outperforms last-layer counterparts in several tasks (ASR, PR, and ER)

**Weaknesses:**

While the experimental results show interesting findings on common ASR, speaker recognition, and emotion benchmarks, these results are not entirely convincing as they rely on relatively simple benchmarks. For ASR evaluation, more challenging robust benchmarks (as used in Whisper or detailed in https://arxiv.org/abs/2401.10446) would be more appropriate. Overall, the experimental validation appears insufficient.

This limitation is reflected in Table 1, where VARAN fails to outperform weighted combination methods - a drawback explicitly mentioned in lines 53-54. This suggests that VARAN may still 'not account for sample variation and its possible dependence on information from SSL layers.'

The reliance on human knowledge for prior distribution is problematic. Specifically, finding appropriate prior distributions for each task is impractical in real-world applications. Even within a single task like ASR, it's unclear whether a single prior distribution can effectively handle variations across languages, dialects, and noise conditions. This raises questions about the method's practical utility, given that humans already inherently adjust such 'priors' for different tasks.

Several technical details require clarification:

How is discrete variational inference (both prior and posterior) modeled? Is Gumbel-Softmax used? The paper mentions softmax application but lacks implementation details.

The architecture appears to differ between speaker-dependent and speaker-independent tasks - this needs elaboration.
State-of-the-art ASR systems typically incorporate language model integration. Why was this omitted?

In conclusion, while the core idea is interesting, the experimental validation is insufficient to fully support the proposed approach.

**Questions:**

See Weaknesses

---

> ### Author Response · Authors · 2024-11-17
>
> Dear Reviewer,
>
> Thank you for your time and for providing a detailed review. We appreciate your mention of the paper's clarity and conciseness and your recognition that VARAN adopts variational inference to semi-automatically extract layer-wise information, providing novel insights.
>
> We noticed that you mentioned the PR task; however, we would like to clarify that there is no such task in our experiments section. For more details, please refer to Section 4.
>
> Regarding the weaknesses you pointed out and your questions:
>
> 1. **How is discrete variational inference (both prior and posterior) modeled? Is Gumbel-Softmax used?**
>     - The prior distribution is modeled using the distribution's parameters (e.g., \(p\) for the geometric distribution); note that this distribution is not learnable. The posterior distribution is modeled via the MHSA block (Figure 1). The input to the MHSA block consists of hidden states from the Transformer Encoder block. Formally, this is a list with a size equal to the number of layers, where each element is a tensor with dimensions [batch size, sequence length, hidden dimension]. We then apply mean pooling along the sequence length and concatenate the results into a single tensor, resulting in dimensions [batch size, number of layers, hidden dimension]. This tensor serves as input to the standard MHSA mechanism, functioning as a feature mixer between layers. Finally, we apply a linear layer to the last dimension to obtain a tensor with dimensions [batch size, number of layers, 1], and softmax is applied along the number of layers dimension to produce an individual posterior distribution for each sample in the batch. A detailed description will be provided in Section 3.2.
> 2. **VARAN Evaluation on ASR Using More Complex Benchmarks**:
>     - We intentionally did not use a language model in our ASR task evaluation setup. Our goal is to explore the best utilization of SSL model layers, and comparing layer-aggregation methods after LM-decoding could introduce bias. While we agree that incorporating strong models into the evaluation pipeline can improve GER and minimize differences between layer-aggregation methods in terms of WER score, we believe this does not discredit the method overall.
> 3. **The architecture appears to differ between speaker-dependent and speaker-independent tasks - this needs  elaboration:**
>     - We trained all models in a speaker-independent setup. If we misunderstood your question, please let us know so we can address it more accurately.
> 4. **State-of-the-art ASR systems typically incorporate language model integration. Why was this omitted?**
>     - In this paper, we do not propose a state-of-the-art ASR system but rather a layer-aggregation method applicable to various SSL architectures and downstream tasks. As mentioned previously, in the experimental setup, we omit rescoring SSL model predictions with the Language Model to avoid introducing bias in the comparison of layer-aggregation methods.
> 5. **The reliance on human knowledge for prior distribution is problematic. Specifically, finding appropriate prior distributions for each task is impractical in real-world applications:**
>     - As mentioned in Section 3.3, for most tasks where we do not have any prior knowledge of layer importance, we suggest selecting the best prior distribution using a hyperparameter search of the chi-squared distribution, as it can be easily adapted to most tasks. Please refer to Section 3.2 and Appendix B for more details.
> 6. **Even within a single task like ASR, it's unclear whether a single prior distribution can effectively handle variations across languages, dialects, and noise conditions:**
>     - As mentioned above, the prior distribution is parameterized by the parameters of a statistical distribution, which can be considered a hyperparameter of the model. Thus, it should not be generalized for multiple languages, dialects, and noise conditions. Like other hyperparameters, such as learning rate or batch size, it should be chosen individually based on the validation set.

---

> > ### Author Response · Authors · 2024-11-17
> >
> > 7. **VARAN underperformance to weighted sum on ASR (test-other) suggests that VARAN may still not account for sample variation and its possible dependence on information from SSL layers:**
> >     -  The sample dependence on information from SSL layers is derived from the use of Variational Inference in the task formulation, as explained in Section 3.1. For more details, please refer to Appendix A. Note that in this paper, we propose a general layer aggregation method and conduct experiments not only with ASR but with three tasks, where VARAN outperforms other methods in tasks requiring both low-level features (such as speaker verification) and high-level features (such as speech or emotion recognition), as shown in Tables 2 and 3. Regarding the ASR task, VARAN still outperforms other methods on the test-clean set but not on the test-other set. We suggest this might be because the model was fine-tuned on the LibriSpeech 100-clean training set, and the test-other set might be considered an out-of-distribution dataset.
> >
> > Thank you once again for your valuable feedback.

---

> > > ### Comment · Reviewer_YerL · 2024-11-25
> > > **Response to Rebuttal**
> > >
> > > I thank the authors for detailed responses. However, issues remain.
> > >
> > > 1. The MHSA model's approach to single-value prediction does not adequately address label independence and value dependence. For instance, while the numerical values between layer 1 and layer 10 show large gaps, these should be treated as two distinct classes rather than continuous values. The authors should review and reference existing work on discrete posterior modeling, such as Gumbel softmax, which could serve as a strong starting point.
> > > 2.    Evaluating the method solely on relatively simple ASR benchmarks like LibriSpeech is insufficient to demonstrate its effectiveness, independent of language model integration. Additionally, my previous concern about generalizability across different languages and dialects remains unaddressed, a limitation the authors have acknowledged. While VARAN shows promise in speech representation learning, comprehensive evaluation on more challenging datasets is needed to verify its capabilities.
> > >
> > > Due to these concerns regarding posterior modeling (lacking both explanation and relevant references) and limited evaluation of generalization capabilities, this paper could benefit from a substantial revision. Hence, I maintain my original score for the paper.

---

> > > > ### Author Response · Authors · 2024-11-26
> > > >
> > > > Following our previous reply, we would like to inform you that we have uploaded a revised version of the manuscript, which we believe provides additional justification for some of your concerns.
> > > >
> > > > > Evaluating the method solely on relatively simple ASR benchmarks like LibriSpeech is insufficient to demonstrate its effectiveness…
> > > > >
> > > >
> > > > We have updated our work to include several additional results for the ASR task.
> > > >
> > > > - We added the Data2Vec Large backbone to our LibriSpeech experiment, and VARAN consistently outperforms other methods (See Table 1).
> > > > - We also added an experiment with ASR for the Russian language, as it was not included in the WavLM training data (See Table 16).
> > > >
> > > > We demonstrate the effectiveness of our method across a diverse range of tasks (not solely on relatively simple ASR benchmark), datasets, and backbone models, including:
> > > >
> > > > - **ASR**:
> > > >     - LibriSpeech, using two different backbone models
> > > >     - CommonVoice, using the WavLM backbone
> > > > - **SER**:
> > > >     - IEMOCAP, using two backbone models
> > > >     - RAVDESS, using two backbone models
> > > > - **SV**:
> > > >     - VoxCeleb1, using the WavLM backbone
> > > >
> > > > We believe that this comprehensive set of experiments, using various tasks and backbones, sufficiently showcases the effectiveness of our method.
> > > >
> > > > We highlighted all changes in red for your convenience. If you have any questions, we will be happy to answer them.

---

> ### Author Response · Authors · 2024-11-25
>
> Thank you for your thoughtful response! We would like to provide some clarifications to ensure that we are on the same page.
> > For instance, while the numerical values between layer 1 and layer 10 show large gaps, these should be treated as two distinct classes rather than continuous values.
>
>
> Parametrization of our posterior distribution allows us to handle this dependency (gradients from the posterior flow through the entire model, except the feature extractor, so that all model parameters are learned to provide a better distribution). If there is no need to apply weights to the distanced layers, the model will learn it.
>
> **The ability of the model to allocate weights to distant layers is a strength, not a flaw**. An example can be imagined where both low and high-level features are needed to correctly predict a label (e.g. emotion recognition depends on text and intonation).
> > such as Gumbel softmax
>
>
> We believe Gumbel softmax is used for other cases where you need to output one value from your discrete distribution during inference, allowing us to obtain a differential approximation of the argmax. In our case, we combine our predictions with continuous weight from posterior PDF output during inference. Gumbel softmax also introduces additional variance and yields loosen lower bound during training.
>
> Let us also note that Gumbel softmax still requires logits as input, so parameterizing the logits is completely independent of using Gumbel. One could use MHSA, RNN, or even a linear layer to obtain logits. Therefore, there is no reason that the probability distribution obtained after Gumbel softmax will have only one mode.
> > Evaluating the method solely on relatively simple ASR benchmarks like LibriSpeech is insufficient to demonstrate its effectiveness
>
>
> We are currently working on the additional results and hope to get it in a 2 days.
>
> Let us note that we believe the goal of our paper is to present a novel method for feature aggregation for a wide variety of tasks, rather than a SoTA ASR model.
>
> If you have any questions or need further clarification, we are more than happy to answer.

---

> ### Author Response · Authors · 2024-11-28
> **Kindly Reminder**
>
> Dear Reviewer "YerL",
>
> We would like to remind you that we have updated our paper since your last review, and we believe the experiments we have provided address your concerns. In light of these revisions, we kindly ask you to reconsider your score. If you have any further questions, we would be happy to address them.

---

> > ### Author Response · Authors · 2024-12-02
> >
> > Dear Reviewer "YerL",
> >
> > We would like to remind you that we have updated our paper since your last review, and we believe the experiments we have provided address your concerns. In light of these revisions, we kindly ask you to reconsider your score. If you have any further questions, we would be happy to address them.

---

### Official Review · Reviewer_dJSB · 2024-11-06

**Soundness:** 3
**Presentation:** 3
**Contribution:** 2
**Rating:** 6
**Confidence:** 5

**Summary:**

This paper introduces VARAN, an innovative layer aggregation technique for finetuning self-supervised speech models motivated from variational inference field. The authors emphasize that adaptively adjusting the weights of outputs from various model layers based on individual samples is essential for boosting performance in downstream tasks. VARAN generates weights for each layer tailored to the specific sample, achieving superior results compared to existing layer-aggregation methods like weighted sum and layerwise attention pooling (e.g., SV and SER). Additionally, it demonstrates competitive performance on the ASR task.

**Strengths:**

VARAN adapts the weight of different layers used conditioned on each sample. It is shown to be very effective for the SER task.

**Weaknesses:**

The authors do not compare VARAN to adapters, which are (i) a more cost-effective finetuning method (involving fewer parameters) and (ii) better suited for adapting models to different downstream tasks, especially when there is limited data available for finetuning. This is demonstrated in the paper, "CHAPTER: Exploiting Convolutional Neural Network Adapters for Self-Supervised Speech Models."

For tasks like ASR, SV, and SER, the performance gains with VARAN may not be as significant compared to adapters, as shown in the CHAPTER paper. Thus, a direct comparison becomes essential to substantiate general claims about VARAN.

Moreover, the reliance on choosing an appropriate prior for each task underscores the importance of understanding what each layer encodes. Tables 4 and 5 demonstrate that selecting the right prior is crucial for achieving performance gains.

**Questions:**

In Figure 1, how multihead self attention is used is unclear to me.

Why using data2vec for SER task and not for all the tasks?

Typo in line 34. with these classical >> with classical.

---

> ### Author Response · Authors · 2024-11-17
>
> Dear Reviewer,
>
> Thank you for your time and for providing a detailed review. We appreciate your recognition of VARAN's ability to adapt layer weights for each individual sample and are glad you found this approach highly effective, especially for the SER task.
>
> Regarding the weaknesses you pointed out and your questions:
>
> 1. **The authors do not compare VARAN to adapters, which are (i) a more cost-effective fine-tuning method (involving fewer parameters) and (ii) better suited for adapting models to different downstream tasks, especially when there is limited data available for fine-tuning:**
>     - While fine-tuning SSL models with adapters [1-3] is indeed more cost-effective, we believe that VARAN should not be directly compared to them, as the methods are orthogonal. VARAN is a layer aggregation method that can be applied to various fine-tuning techniques, as it is an essential step for making predictions. In this paper, we focused on exploring the method itself and demonstrating its application across multiple tasks. Setting up different fine-tuning techniques is beyond the scope of this research; please note that the proposed method can also be applied to fine-tuning with adapters.
>
>         [1] Neil Houlsby and Andrei Giurgiu and Stanislaw Jastrzebski and Bruna Morrone and Quentin de Laroussilhe and Andrea Gesmundo and Mona Attariyan and Sylvain Gelly, Parameter-Efficient Transfer Learning for NLP, ICML, 2019
>
>         [2] Edward J. Hu and Yelong Shen and Phillip Wallis and Zeyuan Allen-Zhu and Yuanzhi Li and Shean Wang and Lu Wang and Weizhu Chen, LoRA: Low-Rank Adaptation of Large Language Models, 2022
>
>         [3] Zih-Ching Chen and Yu-Shun Sung and Hung-yi Lee, CHAPTER: Exploiting Convolutional Neural Network Adapters for Self-supervised Speech Models, ICASSPW, 2022
>
> 2. **Use of Multi-Head Self-Attention (MHSA) for Predicting a Posterior Distribution in Figure 1**:
>     - The input to MHSA block (Figure 1) consists of hidden states from the Transformer Encoder block. Formally, this is a list with a size equal to the number of layers, where each element is a tensor with dimensions [batch size, sequence length, hidden dimension]. We then apply mean pooling along the sequence length and concatenate the results into a single tensor, resulting in dimensions [batch size, number of layers, hidden dimension]. This tensor serves as input to the standard MHSA mechanism, functioning as a feature mixer between layers. Finally, we apply a linear layer to the last dimension to obtain a tensor with dimensions [batch size, number of layers, 1], and softmax is applied along the number of layers dimension to produce an individual posterior distribution for each sample in the batch. A detailed description will be provided in Section 3.2.
> 3. **Why Data2Vec Was Used for the SER Task and Not for All Tasks**:
>     - For a fair comparison of data2vec across all tasks, we needed to run a grid search with optimal hyper-parameters for each task, which is very computationally-intensive. That is why we chose the SER task specifically, as it requires less training time compared to ASR or SV, given that the IEMOCAP dataset is smaller. Despite this limitation, VARAN is designed to work with various upstream models, including wav2vec 2.0 and HuBERT, due to their similar architectures. We believe that the success of the data2vec model utilizing VARAN's layer aggregation method on the SER task demonstrates its generalization ability, but it would be interesting to compare the relative performance improvements on other tasks as well.
>
> During the rebuttal stage, we will strive to expand the experiments section as much as possible, incorporating your comments  about method evaluation.
>
> Thank you once again for your valuable feedback.

---

> > ### Comment · Reviewer_dJSB · 2024-11-25
> >
> > Thank you for the clarification. I have increased the score by one point.

---

> > > ### Author Response · Authors · 2024-11-26
> > >
> > > Thank you for your response, we appreciate that you increased the score!
> > >
> > > Following our initial reply, we would like to inform you that we have uploaded a revised version of the manuscript, which we believe provides additional justification for some of your concerns.
> > >
> > > > Why Data2Vec Was Used for the SER Task and Not for All Tasks
> > >
> > > Please note that we have added results for the Data2Vec model on the ASR task using the LibriSpeech dataset, see Table 1 for more details.
> > >
> > > Additionally, we provide results for ASR using a subset of Common Voice dataset in Russian language and evaluate SER with both WavLM and Data2Vec models on the RAVDESS dataset. These results are present in Appendix F.
> > >
> > > We highlighted all changes in red for your convenience. If you have any questions, we will be happy to answer them.

---

> > > > ### Author Response · Authors · 2024-12-02
> > > >
> > > > Dear Reviewer "dJSB",
> > > >
> > > > We would like to inform you that we have uploaded an updated version of the paper since your last comment. We believe that we have addressed all of your concerns. In light of these revisions, we kindly ask you to review the changes and reconsider your score accordingly.

---

> > > > > ### Comment · Reviewer_dJSB · 2024-12-02
> > > > >
> > > > > Thank you for the response. I still do not see the speech community using this method compared to current SUPERB weighted method. Given the performance gains are not significant and with the additional computations overhead, I will keep my score.

---

> > > > > > ### Author Response · Authors · 2024-12-03
> > > > > >
> > > > > > Thank you for your reply. As we mentioned in the paper, the SUPERB benchmark is designed for upstream model comparison, while VRAN is a layer aggregation method that can be widely used while fine-tuning self-supervised models for a downstream task to achieve a performance gain of up to 20% relative increase over the weighted sum (see Table 17).
> > > > > >
> > > > > > Regarding computational overhead, as shown in Table 8, it is only 4% during inference. In training, 4 hyperparameter assignments are enough for VRAN to outperform baseline methods on SER task.

---

### Author Response · Authors · 2024-11-26
**Short summary of the revised version of the paper**

We thank all reviewers for their valuable feedback!
We believe that during the rebuttal stage, we addressed all the reviewers concerns by providing additional experiments included in the revised paper, which can be summarized as follows:

-  We extended experiments with the Data2Vec model by evaluating it on the second task (ASR); see Table 1 for more details.
- We evaluated VARAN on the ASR task using a subset of the Common Voice dataset in the Russian language to demonstrate VARAN's generalizability for out-of-domain distribution datasets; see Table 16 for more details.
- We added an additional dataset for the SER task and evaluated it on the RAVDESS dataset using both WavLM and Data2Vec models; see Table 17 for more details.
- We reported the number of parameters and latency for the VARAN layer aggregation method in Table 8.
- We provided a detailed explanation of the VARAN layer aggregation method in Section 3.2.
- We have added Appendix D, where we discuss the behavior of the additional hyperparameter of the prior distribution in comparison to baseline hyperparameters. We show that, even with a sufficiently small budget for hyperparameter assignments, our method achieves better performance.

To conclude, in this paper, we proposed a layer aggregation method, VARAN, for fine-tuning speech self-supervised models for different tasks. We demonstrated its superior performance across:

- SER, SV, and ASR tasks;
- WavLM and Data2Vec upstream model backbones;
- LibriSpeech and Common Voice datasets for ASR, and IEMOCAP and RAVDESS datasets for SER.

The proposed method outperforms existing layer aggregation methods in all experimental setups.

---

### Meta-Review · Area_Chair_dkcs · 2024-12-18

**Metareview:**

The paper proposes an alternative to weighted sum of layers for probing. The main strength is to infer the weights per sample instead of using fixed ones. The approach involves predicting the weights on receiving the input and using the weights for downstream.

I recommend rejection because of two reasons: 1) a lack of purpose (i.e., solving the wrong problem) and 2) a lack of details in the writing.

Reviewers dJSB, YerL, and WFfX were confused about the purpose of the paper. The purpose of probing is to see how accessible the a particular type of information is. The weighted sum approach in SUPERB is a proxy to layer-wise probing, and as argued in the paper and in Yang et al. (2024) the weights themselves don't correspond well with the layer performance. It's unclear why we even need to improve the weighted sum approach. As argued in Zaiem et al. (2025), if the goal is to improve downstream, we should opt for bigger probes or (efficient) fine-tuning (e.g., adapters). In fact, the approach in this paper is just one particular type of bigger probes.

Reviewers YerL, 32Z2, and WFfX found it hard to look for details. This is evidenced by the short section 3, despite being the main part of the paper. The reason for doing variational inference is not clear. The paper should at least argue why the likelihood is not tractable.

* Zaiem et al., Speech self-supervised representations benchmarking: a case for larger probing heads, 2025

**Additional Comments On Reviewer Discussion:**

The discussion included a few technicalities, such as the prior in the proposed methods and evaluation, but was mainly about the purpose and utility of the approach. Reviewers are generally not convinced that improving the weighted sum itself is the right direction.

---

### Decision · Program_Chairs · 2025-01-22

Reject